# Genome Mining, Heterologous Expression, Antibacterial and Antioxidant Activities of Lipoamides and Amicoumacins from Compost-Associated *Bacillus subtilis* fmb60

**DOI:** 10.3390/molecules26071892

**Published:** 2021-03-26

**Authors:** Jie Yang, Qingzheng Zhu, Feng Xu, Ming Yang, Hechao Du, Xiaoying Bian, Zhaoxin Lu, Yingjian Lu, Fengxia Lu

**Affiliations:** 1Jiangsu Key Laboratory of Marine Bioresources and Environment, Jiangsu Ocean University, Lianyungang 222005, China; yangjie0737@163.com (J.Y.); zhuqingzheng2020@163.com (Q.Z.); xufeng0737@163.com (F.X.); 2Co-Innovation Center of Jiangsu Marine Bio-industry Technology, Jiangsu Ocean University, Lianyungang 222005, China; 3Jiangsu Marine Resources Development Research Institute, Lianyungang 222000, China; 4Helmholtz Institute of Biotechnology, State Key Laboratory of Microbial Technology, Shandong University, Qingdao 266237, China; yangming_232514@163.com (M.Y.); bianxiaoying@sdu.edu.cn (X.B.); 5College of Food Science and Technology, Nanjing Agricultural University, 1 Weigang, Nanjing 210095, China; duhechao1990@163.com (H.D.); fmb@njau.edu.cn (Z.L.); 6College of Food Science and Engineering, Nanjing University of Finance and Economics, Nanjing 210003, China

**Keywords:** *Bacillus subtilis*, NRPS**/**PKS, amicoumacins, heterologous expression, bioactivities

## Abstract

*Bacillus subtilis* fmb60, which has broad-spectrum antimicrobial activities, was isolated from plant straw compost. A hybrid NRPS/PKS cluster was screened from the genome. Sixteen secondary metabolites produced by the gene cluster were isolated and identified using LC-HRMS and NMR. Three lipoamides D–F (**1**–**3**) and two amicoumacin derivatives, amicoumacins D, E (**4**, **5**), were identified, and are reported here for the first time. Lipoamides D–F exhibited strong antibacterial activities against harmful foodborne bacteria, with the MIC ranging from 6.25 to 25 µg/mL. Amicoumacin E scavenged 38.8% of ABTS^+^ radicals at 1 mg/mL. Direct cloning and heterologous expression of the NRPS/PKS and *ace* gene cluster identified its importance for the biosynthesis of amicoumacins. This study demonstrated that there is a high potential for biocontrol utilization of *B. subtilis* fmb60, and genome mining for clusters of secondary metabolites of *B. subtilis* fmb60 has revealed a greater biosynthetic potential for the production of novel natural products than previously anticipated.

## 1. Introduction

*Bacillus subtilis* is nonpathogenic, and displays considerable genetic diversity, even among closely related strains. Its secondary metabolites have been studied for more than 50 years, and numerous studies have revealed that the secondary metabolites are characterized by antimicrobial and other biological activities. [1] *B. subtilis* has been widely used in the food industry and agriculture for inhibiting and eliminating foodborne and plant pathogens [2,3]. Genomic sequencing revealed that an average of 4–5% of the genome in each strain is devoted to the synthesis of bioactive compounds, giving the organism the potential to produce more than two dozen antibiotics with a great variety of structures [4].

Improvements in sequencing technology have made genome mining an important tool for the discovery of novel natural products [5,6]. During the last decade, the rapid development of bioinformatics tools, as well as improved sequencing and annotation of microbial genomes, has led to the discovery of novel bioactive compounds synthesized by non-ribosomal peptide synthetases (NRPS), polyketide synthases (PKS), and hybrid NRPS/PKS, known as biosynthesis gene clusters (BGCs). This method for the identification of novel substances currently plays an important role in the food industry and in agriculture [7,8,9]. PKS and NRPS synthetic products represent one of the largest classes of microbial natural products, and have important clinical and ecological impacts [10]. PKS and NRPS synthetic products present striking similarities in their biosynthetic assembly mechanisms. The presence of gene clusters allows the existence of hybrid clusters that contain elements of each class, resulting in the formation of hybrid NRPS/PKS [11]. These hybrid NRPS/PKS clusters provide even greater potential for numerous varieties of secondary metabolites to be generated by microorganisms [12,13].

The transfer of BGCs from the original host into a feasible heterologous host, resulting in heterologous expression, has become a valid alternative approach for identifying the gene clusters required for the biosynthesis of compounds. In 2012, we described a method termed linear plus linear homologous recombination (LLHR) to directly clone large BGCs from digested genomic DNA in *Escherichia coli*. LLHR is mediated by the prophage recombinase RecET in *E. coli* [14]. In 2016, an improved method, exonuclease combined with RecET recombination, was described [15].

In the present study, wild-type *B. subtilis* fmb60 isolated from compost was shown to exhibit a broad spectrum of antimicrobial activities. The genome sequence of *B. subtilis* fmb60 was investigated, and potential secondary metabolite clusters were predicted using antiSMASH. The genome contained three NRPS clusters, one type I PKS cluster, and one NRPS/PKS cluster. Several products of the NRPS cluster metabolites have already been confirmed, including surfactin, fengycin, and bacillibactin. The type I PKS cluster metabolites have also been identified in our previous studies, involving aurantinins B, C, and D [16]. However, the secondary metabolites synthesized by the hybrid NRPS/PKS cluster of *B. subtilis* fmb60 have not been well studied. In this study, based on the analysis of hybrid NRPS/PKS clusters, three new lipoamides D–F (**1**–**3**), two new amicoumacins D, E (**4**, **5**), and 11 known amicoumacins metabolites (**6**−**16**) were identified in this study. The antimicrobial activity of **1**–**16** was evaluated. Compounds **1**–**3**, **7**, and **11** showed significant antibacterial activity. The antioxidant activity of **6**–**16** was evaluated. Compounds **5**, **6**, **13**, **15**, and **16** showed significant FRAP activities and ABTS^+^ radical scavenging activities. The ExoCET method was used to directly clone and synthesize related gene clusters and perform metabolite analysis on *B. subtilis* fmb60.

## 2. Results

### 2.1. Genome Sequencing, Annotation, and Bioinformatics Analysis

The genome of *B. subtilis* fmb60 was sequenced, and three scaffolds were acquired. The draft genome sequences were deposited in GenBank under accession number LYMC01000002.1. Using the gene prediction software Glimmer, 3045 genes were predicted, of which 2829 genes were identified as having homologues. These homologous proteins were derived from 80 species, including *Bacillus* sp. JS, *B. subtilis* subsp. *subtilis* str. 168, and *B. subtilis* QB928. *Bacillus* sp. JS contained the greatest number of homologous proteins, accounting for 63.6% [17].

Analysis of the *B. subtilis* fmb60 genome using antiSMASH 5.2.0 revealed 17 possible secondary metabolite biosynthetic gene clusters, of which three NRPS clusters, two PKS clusters, and one hybrid NRPS/PKS cluster had the capacity to produce bioactive metabolites. Focusing on the gene cluster of the hybrid NRPS/PKS, it was found that this gene cluster was homologous with the genome of *B. subtilis* subsp. *inaquosorum* KCTC 13,429 (91%) and the biosynthetic genes of xenocoumcin (28%). Xenocoumacin includes Xcn 1 and Xcn 2, which are amicoumacin derivatives with amino acid and carboxylic acid hybrid moieties. These derivatives are the major antimicrobial compounds isolated from *Xenorhabdus nematophila* [18]. Further studies showed that a hybrid NRPS/PKS gene cluster (*xcn*A-*N*) was required for their synthesis [19]. The *xcnA* and *xcnG* genes are responsible for synthesizing an important precursor (*N*-acyl-d-Asn), and cleaving prexenocoumacins in the production of xenocoumacin in the biosynthesis gene cluster of xenocoumacin. Thus, the hybrid NRPS/PKS cluster from *B. subtilis* fmb60 was predicted to synthesize amicoumacins.

### 2.2. Isolation and Characterization of Lipoamides

In searching for possible lipoamides, it was noticed that apart from fengycin, surfactin, and bacillibactin, another fraction of the crude extracts was eluted from the semi-preparation HPLC. Three novel compounds were isolated from this fraction.

The structure of compound **1** was determined using ^1^H and ^13^CNMR spectroscopy and heteronuclear single quantum coherence spectroscopy (HSQC) data. The ^13^C NMR and DEPT spectra displayed 35 carbon signals, classified as three carbonyls, two methyl, 10 methylene, and two methylidynes (Table 1). The planar structure of compound **1** was established using two-dimensional (2D) NMR data (Figure 1A). In the HSQC, ^1^H−^1^H COSY, and heteronuclear multiple bond correlation (HMBC) spectra of compound **1**, C-2’ (δ_C_ 50.5, δ_H_ 4.70, t) was easily recognized and set as a starting point. The key HMBC correlations of H-2′ (δ_H_ 4.70, t) to C-1′ (δ_C_ 175.1), C-3′ (δ_C_ 37.9), C-4′ (δ_C_ 174.1), and C-1 (δ_C_ 176.2), as well as H-3′ (δ_H_ 2.75, 1.31, m) to C-1′ (δ_C_ 175.1) and C-4′ (δ_C_ 174.1) indicated the existence of an Asn unit. Two methyl proton signals at δ_H_ 0.85 and a broad peak at δ_H_ 1.31 indicated the presence of an *iso* fatty acid side chain, a finding which was supported by the correlation spectroscopy (COSY, HSQC, and DEPT) data. Two ^1^H−^1^H COSY spin subunits were recognized. These two subunits are linked through an amide bond, as evidenced by the HMBC correlation from H-2′ (δ_H_ 4.70, t) to C-1 (δ_C_ 176.2). Further comprehensive analysis of 2D NMR data showed that the branched-chain methyl fatty acid of compound **1** is different from that of lipoamide A and other similar structures [20,21]. The position of the methyl group at C-13, which was manifested by the HMBC correlations of C-13 (19.7) to H-11 (δ_H_ 1.31, 1.09), and C-12(δ_C_ 11.8) was correlated with H-11 (δ_H_ 1.31, 1.09). Thus, the structure of compound **2** was proposed, as shown in Figure 1B, and named lipoamide D.

The structure of compound **2** was investigated using NMR spectroscopy. These data revealed that compound **2** possessed structural similarities to compound **1,** but differed from the molecular formula of compound **1** by the addition of –CH_2_ (δ_C_ 30.4, δ_H_ 1.31) (Table 1). By analyzing the HSQC and HMBC, the position of the methyl group was confirmed at C-12. Thus, compound **2′**s structure was proposed, as shown in Figure 1B, and it is first named lipoamide E.

The structure of compound **3** was investigated using NMR spectroscopy. The 1D NMR spectra of compound **3** exhibited high similarities to those of compound **1**, indicating that its structure was closely related to that of compound **1** (Table 1). The only difference was that the molecular weight of compound **3** was higher than 14 amu of compound **1**, while an additional carbon resonance signal (δ_C_ 30.3) could be observed. Meanwhile, the position of the methyl group was confirmed at C-14 according to HSQC and HMBC analysis. Thus, compound **3′**s structure was proposed, as shown in Figure 1B, and it is named lipoamide F for the first time.

### 2.3. Isolation and Characterization of New Amicoumacins

The fermentation broth of strain *B. subtilis* fmb60 was extracted with ethyl acetate. Thirteen compounds were isolated using HPLC, all of which exhibited similar UV spectra, indicating that they were probably structurally related. Analysis of the ^1^H, ^13^C NMR, and ESI-HRMS/MS spectra of this extract revealed that the structure of the bioactive fractions was almost identical to that of isocoumarin-type compounds [22].

Compound **4** was isolated as an amorphous solid. The ESI-MS/MS spectrum of the precursor ion [M + H]^+^ was dominated by common ion peaks at *m/z* 372.1, 250.1, 233.1, and 215.1 (Figure 2A). The ^1^H and ^13^C NMR resonances (Table 2) as well as ^1^H−^1^H COSY and HSQC, suggested the presence of 21 carbons, including three carbonyl carbons, six aromatic carbons, four methylenes, a methine, two methyls, and five carbons bonded to nitrogen or oxygen. These data revealed that compound **4** possessed structural similarities to amicoumacin C but differed from the molecular formula of amicoumacin C by the addition of –CH_2_ [23]. This methylene group was identified by two proton signals observed at δ_H_ 2.36, 2.23 in the ^1^H NMR spectrum, an observation which was supported by the associated carbon resonating at *δ*_C_ 31.1 in the HSQC spectrum. The ^1^H−^1^H COSY correlations from H-11′ (δ_H_ 2.22, m; 2.16, m) to H-10′ (δ_H_ 3.88, m) and H-12′ (δ_H_ 2.36, m; 2.23, m) suggested that C-11′ (δ_C_ 22.7) was connected with C-10′ (δ_C_ 57.8) and C-12′ (δ_C_ 31.1). The assignment of the methylene group at C-12′ was confirmed by the HMBC correlation from H-11′ (δ_H_ 2.22, m; 2.16, m) and H-12′ (δ_H_ 2.36, 2.23) to carbonyl carbon C-13′ (δ_C_ 181.5). The HMBC correlation of H-10′ (δ_H_ 3.88, m) and C-13′ (δ_C_ 181.5) suggested the presence of an amide bond between C-10′ and C-12′ (Figure 2C). The relative configuration of compound **4** was determined by NOESY correlations of H-3/H-4′, H-5′/H-8′, and H-8′/H-10′, and shares stereochemical configurations similar to those of amicoumacin C. Thus, compound **4** was assigned as a new amicoumacin derivative from the spectroscopic data analysis, and was named amicoumacin D.

Compound **5** was isolated as an amorphous solid. The ESI-MS/MS spectrum of the precursor ion [M + H]^+^ was dominated by common ion peaks at *m/z* 372.1, 250.1, 233.1, and 215.1 (Figure 2B). The 1D NMR data (^1^H and ^13^C NMR, Table 2) and 2D NMR (^1^H-^1^H COSY and HSQC data) suggested the presence of 26 carbons, including four carbonyl carbons, six aromatic carbons, two methylenes, two methines, five methyls, and seven carbons bonded to nitrogen or oxygen. These data revealed that compound **5** possessed structural similarities to the 14′, 15′-methyl-15′-hydroxypyrrolidine amicoumacin C, but differed from the molecular formula of 14′, 15′-methyl-15′-hydroxypyrrolidine amicoumacin C by the addition of –COCH_3_ [21]. This acetyl group was identified by a distinct singlet peak observed at δ_H_ 2.05 in the ^1^H NMR spectrum and a carbonyl carbon (δ_C_ 172.1) in the ^13^C NMR spectrum, which was supported by the HMBC correlation between H-17′ (δ_H_ 2.05, s) and carbonyl carbon C-16′ (δ_C_ 172.1). The ^1^H−^1^H COSY correlations from H-13′ (δ_H_ 1.19, d, 5.5) to H-14′ (δ_H_ 3.76, dd, 5.5, 11.0), and HMBC correlations of H-13′/H-14′ and HMBC correlations from H-13′ (δ_H_ 1.19) and H-18′ (δ_H_ 1.44) toC-15′ (δ_C_ 78.1) suggested that the acetyl group was positioned at C-15′ (Figure 2C). The relative configuration of compound **5** was determined by NOESY correlations of H-3/H-4′, H-5′/H-8′, H-8′/H-11′, H-11′/H-18′, and H-14′/H-18, which shared similar stereochemical configurations with 14′, 15′-methyl-15′-hydroxypyrrolidine amicoumacin C. Thus, compound **5** was also assigned as a new amicoumacin derivative, and was named amicoumacin E for the first time.

Comparison of the 1D and 2D NMR data with the references confirmed that the structures of some compounds were the same as some known compounds, such as N-butanonyl-amicoumacin C, **6**, amicoumacin A–C, **7**–**9**, bacilosarcin A–C, **10**–**12**, N-acetylamicoumacin B, C, **13**, **14**, AI-77-H, **15**, and lipoamicoumacins B, **16** [20,22,23,24,25]. Compound **6** was identified as a new natural amicoumacin, which is referred to herein as N-butanonyl-amicoumacin C.

### 2.4. Direct Cloning and Heterologous Expression of NRPS/PKS Gene Cluster

To connect the NRPS/PKS gene cluster with amicoumacin biosynthesis, we direct cloned these DNA regions into *E. coli* plasmids using ExoCET and LLHR. Using the cloning vector p15A-*cm*-*tet*^R^-*ccdB*-*hyg* as the template, we designed a pair of primers carrying the terminal 62-bp homology arms of the NRPS/PKS gene cluster. The 5′ and 3′ homology arms were selected at the locations of the restriction enzymes site *Spe*I and *Pac*I flanking the gene cluster. Using ExoCET, this biosynthetic gene cluster was successfully cloned into the cloning vector, and the efficiency 2/13 verified by restriction enzyme analysis (Appendix A). The resulting recombinant plasmids containing the NRPS/PKS gene cluster were designated as p15A-*cm*-*ami*. Consistent with the close genetic relationship between the native host and the heterologous host, the NRPS/PKS gene cluster was under the control of its original promoters.

The recombinant plasmid p15A-*cm*-*ami* was transferred into *E. coli* GB05-MtaA by electroporation, and verified by restriction enzyme analysis, to obtain *E. coli* GB05-MtaA-*ami*. The correct transformants were subsequently fermented, and crude extracts were analyzed using HPLC-MS to detect the products. The resulting transformants successfully produced two amicoumacins—amicoumacin A (*m/z* 424.2006 [M + H]^+^) and amicoumacin C (*m/z* 407.17 [M + H]^+^—at levels 100-fold less than that in the native *B. subtilis* fmb60 strain (Figure 3A). This result indicated that the NRPS/PKS locus encodes amicoumacin biosynthesis. However, only two amicoumacins were obtained, and the biosynthetic pathways of all other amicoumacins require further investigation.

### 2.5. Direct Cloning and Heterologous Expression of NRPS/PKS Gene Cluster and ace Gene Cluster

Using the cloning vector pBR322-*apra*-OriT and genomic DNA as the template, we designed two pairs of primers to amplify the pBR322-*apra* cloning vector and the *ace* gene cluster. The resulting recombinant plasmids containing the *ace* gene cluster were designated as pBR322-*apra*-*ace* (Appendix A). The recombinant plasmid pBR322-*apra*-*ace* was transferred into *E. coli* GB05-MtaA-*ami* by electroporation and verified by restriction enzyme analysis, to obtain *E. coli* GB05-MtaA-*ami-ace* [26]. It was found that after adding the *ace* gene cluster, compared with the NRPS/PKS gene cluster alone, bacilosarcin A (*m/z* 492.20 [M + H]^+^) and N-butanonyl-amicoumacin C (*m/z* 477.20 [M + H]^+^) were identified (Figure 3B). Bacilosarcin B and bacilosarcin A, N-butanonyl-amicoumacin C can be synthesized using the same substrate, indicating that the *ace* gene cluster plays a key role in the biosynthesis of amicoumacins.

### 2.6. Antimicrobial Bioassay of Lipoamides and Amicoumacins

The antimicrobial activities of lipoamides, amicoumacins, and erythromycin gluceptate were evaluated in vitro using a broth microdilution method. As shown in Table 3, lipoamides D–F, amicoumacin A, and bacilosarcin B produced significant inhibition against *S. aureus* ATCC25923, *M. luteus* CMCC 28001, *B. pumilus* CMCC 63202, *B. cereus* ATCC 14579, *L. monocytogenes* CICC 21662, and *S. aureus* MRSA, with MIC values ranging from 1.56 to 25 µg/mL. In particular, the MIC values against *S. aureus* MRSA were 6.25 µg/mL for amicoumacin A and 3.13 µg/mL for bacilosarcin B. However, amicoumacins D, E and other isolated compounds were found to be weakly active or inactive at 100 µg/mL.

### 2.7. Assay of FRAP Activities and ABTS^+^ Radical Scavenging

The antioxidant activities of the amicoumacins were tested according to their FRAP activities and ABTS^+^ radical scavenging assays (Figure 4A,B). Amicoumacin E, *N*-acetylmethyamicoumacin C, *N*-acetylamicoumacin B, AI-77-H, and lipoamicoumacins B showed significant FRAP activities and ABTS^+^ radical scavenging activities at a concentration of 1 mg/mL. In the ABTS^+^ radical scavenging assay, amicoumacin E, *N*-butanonyl-amicoumacin C, *N*-acetylamicoumacin B, AI-77-H, and lipoamicoumacins B scavenged 38.8%, 55.9%, 48.9%, 64.5%, and 69.9% of ABTS^+^ radicals, respectively. Other amicoumacins had limited antioxidant activities at the same concentration. Furthermore, the antioxidant activities of all amicoumacins were weaker than those of Trolox (lipoamides not tested).

## 3. Discussion

*B. subtilis* has significant metabolic capabilities and versatile biochemical mechanisms, because of its production of structurally diverse bioactive chemical structures [27,28]. Recent microarray-based comparative genomic analyses have revealed that members of this species also have diverse genomic properties [29]. There are about 700 *Bacillus* sp. genomes available on the NCBI website. Thus, the genomes presented here can guide the in-depth investigation of other *B. subtilis* metabolites. The genome sequence and analysis of *B. subtilis* fmb60 showed that a variety of clusters were involved in the production of bioactive compounds.

In the present work, five novel lipoamides and amicoumacins were identified from *B. subtilis* fmb60 by genome-directed isolation. Amicoumacins belong to a family of 3, 4-dihydroisocoumarin derivatives produced by the genus *Bacillus*, which have shown antibacterial, anti-inflammatory, and antiulcer activities and potent gastroprotective and antiulcerogenic activities [30,31]. To identify the genetic determinants of lipoamide and amicoumacins biosynthesis in *B. subtilis* fmb60, the NRPS/PKS gene cluster of *B. subtilis* fmb60 genome was submitted to BLAST. Our analysis indicated that the hybrid NRPS/PKS gene cluster was similar to that of the *ami* gene cluster synthase. *ami* was identified in earlier studies as the amicoumacin A biosynthetic gene cluster from *B. subtilis* 1779 [32]. Thus, the hybrid NRPS/PKS gene cluster could be plausibly assigned to the biosynthetic gene cluster for amicoumacins and lipoamides, which were isolated from *B. subtili*s fmb60.

Heterologous expression is a strategy for natural product discovery, and Red/ET recombineering has been widely used for direct cloning of biosynthetic pathways. The ExoCET method was used to direct clone the NRPS/PKS gene cluster of *B. subtilis* fmb60 and subsequently expressed it in the heterologous host *E. coli* GB05-MtaA. Two amicoumacins were obtained, which further verified that NRPS/PKS gene cluster is crucial for the biosynthesis of the amicoumacins.

However, the synthesis mechanism of other amicoumacins has not been reported yet. Previous reports indicated that this may in part be due to the possibility that many of the amicoumacin analogues discussed here may be isolated artifacts [4]. The formation mechanism of the 2-hydroxymorpholine moiety, an unusual cyclic structure in bacilosarcin, cannot be elucidated only by bioinformatics analysis [33]. Previous studies reported that chemical synthesis of bacilosarcins has been accomplished from amicoumacin C, using diastereoselective reductive amination of amicoumacin C with acetoin to form the corresponding *N*-butanonyl-amicoumacin C [21,34]. Therefore, the diastereomeric mixture of *N*-butanonyl-amicoumacin C is the core intermediate in the synthesis of bacilosarcin. In this study, *N*-butanonyl-amicoumacin C was identified in *B. subtilis* fmb60 fermentation broth for the first time. Acetoin is also an important physiological metabolic product excreted from *B. subtilis* [35]. Thus, it is presumed that bacilosarcins from *B. subtilis* are also synthesized using the aforementioned pathway under the catalysis of different enzymes. We used exonuclease combined with ExoCET technology to clone the NRPS/PKS and *ace* gene clusters into *E. coli*. We found that heterologous co-expression strains can synthesize bacilosarcin A and *N*-butanonyl-amicoumacin C, bacilosarcin B, bacilosarcin A, and *N*-butanonyl-amicoumacin C from the same substrate, indicating that the *ace* gene cluster is related to the biosynthesis of bacilosarcins.

For amicoumacin D, Gln was recognized by the *ami*I domain of the NRPS/PKS gene cluster instead of Asn, resulting in the synthesis of bacillmacin A. The synthesis of *N*-acetylamicoumacin A–C from *X. boviensis* was mainly related to *N*-acetyltransferase, and *B. subtilis* also synthesized metabolites with a structure similar to that of *N*-acetylamicoumacin A–C. This study also assumed that *N*-acetyltransferase in *B. subtilis* may take part in the synthesis process [36]. However, focusing on the hybrid NRPS/PKS gene cluster of *B. subtilis* fmb60, no domain related to *N*-acetyltransferase synthesis was found near the genome. Therefore, it is presumed that the synthesis of *N*-acetylamicoumacin B and C in *B. subtilis* may be mediated by the isomerase or through post-modification. Furthermore, the biosynthetic pathways of amicoumacin E still need further investigation.

## 4. Materials and Methods

### 4.1. Bacterial Strains and Culture Conditions

*B. subtilis* fmb60 was isolated from the straw compost of Nanjing. *B. subtilis* fmb60 colonies and transferred to solid medium nutrient agar. *B. subtilis* fmb60 was cultured in beef peptone yeast medium at 37 °C for 16 h as seeds. One milliliter of seed culture was inoculated into 300 mL of Landy medium in a 1 L shake-flask and cultured in a shaking incubator (speed: 180 rpm) at 33 °C for 36 h.

### 4.2. Genome Sequencing, Annotation, and Bioinformatics Analysiszz

The genome of *B. subtilis* fmb60 was sequenced using the third-generation single-molecule sequencing machine PacBio RS II (PacBio, Menlo Park, CA, USA) by Shanghai Hanyu Bio-Tech (Shanghai, China). Total genomic DNA was extracted using bacterial DNA kits (Omega Bio-Tek, USA). The extracted DNA was fragmented to 10 kb using Covaris^®^ g-TUBE^®^ (Covaris, Woburn, MA, USA). Subsequently, the DNA fragment library used as the template for sequencing was constructed using PacBio^®^ SMRTbell™ Template Prep Kits (PacBio, Menlo Park, CA, USA). After sequencing, the fragments were assembled to reconstruct the complete genome of *B. subtilis* fmb60. The bioinformatics program antiSMASH 5.2.0 (Kai Blin, Kgs. Lyngby, Denmark) was initially used to analyze the whole draft genome sequence [37].

### 4.3. Isolation and Purification of Lipoamides and Amicoumacins

To isolate the lipoamides from *B. subtilis* fmb60, the fermentation broth was centrifuged for 15 min at 10,000× *g*. The supernatant was collected and adjusted to pH 2.0 with HCl. After acid precipitation at 4 °C for 12 h, the supernatant was collected by centrifuging at 10,000× *g* for 15 min. Then, the supernatant was extracted twice with the same volume of ethyl acetate, and the solvent was removed under reduced pressure at 50 °C to dryness. The dry matter was dissolved in methanol and centrifuged to collect the supernatant containing the crude substances.

The crude substances generated using the method described above were processed using a semipreparative HPLC system (Waters 600, Milford, MA, USA) to collect fractions from the subfractions. One milliliter of crude extract solution was loaded and separated on a column (150 × 10 mm i.d., 5 μm, C18, Waters; flow rate: 6 mL/min; detection: 210 nm) from different subfractions using an isocratic program with acetonitrile−H_2_O (*v*/*v*) as the eluent. Finally, three novel compounds **1**–**3** were isolated, and ESI-HRMS analysis showed intense pseudo-molecular ions [M + H]^+^ at *m/z* 329.2452, 315.2295, and 343.2604 for these three compounds.

To isolate the amicoumacins from *B. subtilis* fmb60, the acidified supernatant was adjusted to pH 7.0 with NaOH. The supernatant was then extracted with the same volume of ethyl acetate. The solvent was evaporated under reduced pressure at 50 °C to dryness. The dry matter was dissolved in methanol to collect the supernatant containing the crude substances. The crude extract was subjected to Sephadex LH-20 column chromatography (Φ2.6 × 100 cmi.d., Sigma, St. Louis, MO, USA) for further separation. The eluent was MeOH−H_2_O (4:1, *v*/*v*) and the flow rate was 12 mL/h. The eluent fraction was purified by Nexera X2 HPLC (SHIMADZU, Kyoto, Japan) with a gradient mobile phase (acetonitrile−H_2_O, from 50% to 60%; column: 150 × 10 mm i.d., 5 μm, C18, Waters; flow rate: 4 mL/min; detection: 250 nm) to yield compounds **4**–**16**.

### 4.4. Determination of Amicoumacins by HPLC-MS

For liquid chromatography high-resolution mass spectrometry (LC-HRMS) analysis, an UltiMate 3000 HPLC system (Dionex, Sunnyvale, CA, USA) equipped with a C18 column (5 µm, 4.6 × 250 mm i.d., Agilent Technologies, Palo Alto, CA, USA) was used. The spray voltage was set to 4.0 kV, and the heated transfer capillary temperature was set to 350 °C. High-resolution mass spectrometry analysis was performed with a Thermo Finnigan Surveyor-LCQ DECA XP Plus (Thermo Electron Corporation, San Jose, CA, USA) equipped with an electrospray ionization (ESI) source.

### 4.5. NMR Spectroscopy Identification

The structures were identified by nuclear magnetic resonance (NMR) spectroscopy, and the samples were carried out in methanol-*d*_4_ on Avance 600, 500, and 300 MHz Bruker NMR spectrometer (Bruker, Faellanden, Switzerland).

*Lipoamide D* (**1**): amorphous solid, UV (MeOH) *λ*_max_ (log *ε*) 210 (2.10), 248 (0.53), 315 (0.39) nm, HR-ESI-MS *m/z* 329.2452 [M + H]^+^ (Calcd. 329.2435 C_17_H_33_N_2_O_4_ [M + H]^+^); Soluble, methanol, DMSO, CHCl_3_, ethyl acetate; insoluble/poorly soluble in water.

*Lipoamide E* (**2**): amorphous solid, UV (MeOH) *λ*_max_ (log *ε*) 210 (2.08), 248 (0.51), 315 (0.37) nm, HR-ESI-MS *m/z* 315.2295 [M + H]^+^ (Calcd. 315.2278 C_16_H_31_N_2_O_4_ [M + H]^+^); Soluble, methanol, DMSO, CHCl_3_, ethyl acetate; insoluble/poorly soluble in water.

*Lipoamide F* (**3**): amorphous solid, UV (MeOH) *λ*_max_ (log *ε*) 210 (2.10), 248 (0.55), 315 (0.40) nm, HR-ESI-MS *m/z* [M + H]^+^ 343.2604 (Calcd. 343.2591 C_18_H_35_N_2_O_4_ [M + H]^+^); Soluble, methanol, DMSO, CHCl_3_, ethyl acetate; insoluble/poorly soluble in water.

*Amicoumacin D* (**4**): amorphous solid, [α]^23^_D_−93.2° (*c* 0.1, MeOH); UV (MeOH) *λ*_max_ (log *ε*) 203 (4.43), 246 (3.74), 314 (3.54) nm, HR-ESI-MS *m/z* 421.1971 [M + H]^+^ (Calcd. 421.1975 C_21_H_29_N_2_O_7_ [M + H]^+^); Soluble, methanol, DMSO, CHCl_3_, ethyl acetate; insoluble/poorly soluble in water.

*Amicoumacin E* (**5**): amorphous solid, [α]^23^_D_−83.7° (*c* 0.1, MeOH); UV (MeOH) *λ*_max_ (log *ε*) 203 (4.42), 246 (3.70), 314 (3.44) nm, HR-ESI-MS *m/z* 519.2348 [M + H]^+^ (calcd. 519.2343 C_26_H_35_N_2_O_9_ [M + H]^+^); soluble, methanol, DMSO, CHCl_3_, ethyl acetate; insoluble/poorly soluble in water.

### 4.6. B. subtilis fmb60 Genomic DNA Isolation

Genomic DNA was isolated from lysed cells by phenol-chloroform extraction and ethanol precipitation [26]. *B. subtilis* fmb60 was cultured in 50 mL of medium LB at 30 °C overnight. After centrifugation at 8300× *g* for 5 min, the cells were washed twice with ddH_2_O, and then resuspended in 8 mL of SET buffer (75 mM NaCl, 25 mM EDTA, 20 mM Tris, pH 8.0). After adding 10 mg lysozyme and incubating at 37 °C for 1–2 h in a water bath with occasional inverting, 500 μL of proteinase K (20 mg/mL) and 1 mL of 10% sodium dodecyl sulfate (SDS) were added, and the mixture was incubated at 50 °C with occasional inversion for 2 h until the solution became clear. Then, 3.5 mL of 5 M NaCl was added with inverting, after which 15 mL of phenol/chloroform/isoamyl alcohol (25:24:1) was added and mixed thoroughly by inversion to create an emulsion. After centrifugation at 8300× *g* for 30 min, 500 μL of the aqueous phase was transferred to a new 2 mL tube using an end-cut wide-bore 1 mL tip, then 35 μL of 3 M sodium acetate (pH 7.5) and 1.2 mL of absolute ethanol were added, and gently inverted to mix. The DNA was transferred to a 1.5 mL tube using a blue tip, rinsed with 75% ethanol, dried at room temperature (RT) for approximately 30 min, and dissolved with 200 μL ddH_2_O. Genomic DNA was prepared.

### 4.7. Direct Cloning of the NRPS/PKS and ace Gene Clusters

*E. coli* were cultivated and manipulated according to standard protocols. The strains and plasmids used in this study are listed in Appendix A. Primer synthesis and DNA sequencing were performed at Shanghai Sangon Biotech Co., Ltd. (Shanghai, China). Restriction enzymes were purchased from New England Biolabs Ltd. (Beijing, China), DNA polymerase (PrimerSTAR and T4), and DNA marker were purchased from TAKARA Biotechnology Co., Ltd. (Dalian, China).

The genomic DNA was completely digested with the restriction enzymes *Pac*I and *Spe*I to release the NRPS/PKS gene cluster. The linear cloning vector p15A-cm flanked with homology arms to target genes was amplified by PCR using p15A-*cm*-*tet*^R^-*ccdB*-*hyg* as template, and primers (YM-*ami*-5 and YM-*ami*-3, Appendix A) carrying the terminal homology arms to the target gene cluster. A direct cloning protocol was carried out according to our previous publication [26]. Recombinants carrying the entire gene cluster were selected by chloramphenicol resistance on the plate (the final concentration was 15 μg/mL) and subsequently verified by restriction enzyme analysis and sequencing, to obtain correct recombinants.

The *ace* gene cluster was amplified by PCR using primers *ace*-F and *ace-*R carrying the terminal homology arms to the pBR322 cloning vector. The linear cloning vector pBR322-apra flanked with homology arms to the *ace* gene cluster was amplified by PCR using pBR322-*apra-*OriT as template and primers (*ace*-pBR322-HAF and ace-pBR322-HAF, Appendix A) carrying the terminal homology arms to the *ace* gene cluster. These two fragments were recombined using RecET recombinase in strain *E. coli* GB05-dir. The protocol was carried out according to our previous publication [14]. Recombinants carrying the entire gene cluster were selected by apramycin resistance on the plate (the final concentration was 50 μg/mL) and subsequently verified by restriction enzyme analysis and sequencing, to obtain correct recombinants.

### 4.8. Heterologous Expression of NRPS/PKS and ace Gene Clusters in E. coli

The recombinant plasmid p15A-*cm-ami* was transferred into *E. coli* GB05-MtaA by electroporation and verified by restriction enzyme analysis to obtain *E. coli* GB05-MtaA-*ami* [27]. The plasmid pBR322-apra-*ace* was transferred into *E. coli* GB05-MtaA-*ami* by electroporation to obtain recombinant *E. coli* GB05-MtaA-*ami-ace* containing the NRPS/PKS and ace gene clusters.

The correct transformants were subsequently fermented. The fermentation method for *B. subtilis* fmb60 is described above, while *E. coli* GB05-MtaA-*ami* and *E. coli* GB05-MtaA-*ami-ace* were cultured in beef peptone yeast medium at 37 °C overnight as seeds. One milliliter of seed culture was inoculated into a 250 mL shake-flask containing 50 mL of Landy medium with agitation at 200 rpm at 30 °C for 3 d. The extraction method is described above. The crude extract was obtained using HPLC-MS, which was carried out on a Thermo Scientific UltiMate 3000 HPLC system connected to a Bruker ESI-MS-MS Impact HD operating in positive ionization mode at a scan range of *m/z* 200–2000, auto MS^2^ fragmentation. Reverse-phase chromatography of HPLC was carried out with 2.1 × 100 mm, 2.2 μm columns (PSLC 120 C18) in a solvent gradient, with solvents A (water and 0.1% formic acid) and B (CH3CN and 0.1% formic acid) for 25 min: 5% B from 0 to 3 min, 5% B–95% B within 15 min, followed by 4 min with 95% B and 3 min with 5% B at a flow rate of 0.3 mL/min. MS measurements were carried out using a standard ESI source.

### 4.9. Minimal Inhibitory Concentration (MIC) Assays and the Effects of Antimicrobials on Bacterial Morphology

The MICs of the amicoumacins were determined using a broth microdilution method according to the Clinical and Laboratory Standards Institute (CLSI) guidelines [38]. The indicator bacteria in this assay were incubated in Mueller-Hinton broth (MHB) and normalized to an optical density of 10^6^ CFU/mL. The amicoumacins were two-fold serially diluted in 96-well plates of MHB, with 50 μL in each well and concentrations ranging from 0.78 to 100 μg/mL. Then, 50 μL of bacterial suspension was added to each well, with a final inoculum density of approximately 5 × 10^5^ CFU/mL, and the plates were incubated statically at 37 °C for 24 h [39]. Quality control and interpretations of results were performed according to CLSI guidelines [40]. The MICs of the amicoumacins were defined as the lowest concentration of amicoumacins, where no visible growth was observed in the wells of the microtiter plates. All assays were performed in triplicate.

### 4.10. Assay of ABTS^+^ Radical Scavenging and Ferric-Reducing Activity

The ferric-reducing ability power (FRAP) and ABTS^+^ radical scavenging activities of the amicoumacins were measured using commercial kits (Jiancheng Bio-engineering Institute, Nanjing, China). Trolox was used as a positive control.

### 4.11. Data Processing

SPSS 17.0 software was used for the experimental data processing. Antibacterial and antioxidant experiments were repeated three times, and the Duncan test was used for significance analysis (*p* < 0.05). All of the forms were made using Excel 2010.

## 5. Conclusions

In conclusion, five new metabolites, lipoamides D–F (**1**–**3**), amicoumacins D, E (**4**, **5**), and known metabolites (**6**−**16**), have been identified from *B. subtilis* fmb60. Lipoamides D–F inhibited the growth of foodborne harmful bacteria, and amicoumacins D and E have antioxidant activities. The heterologous expression results demonstrated the function of NRPS/PKS in *B. subtilis* fmb60, and showed that the *ace* gene cluster can participate in the synthesis of amicoumacins.

## Figures and Tables

**Figure 1 molecules-26-01892-f001:**
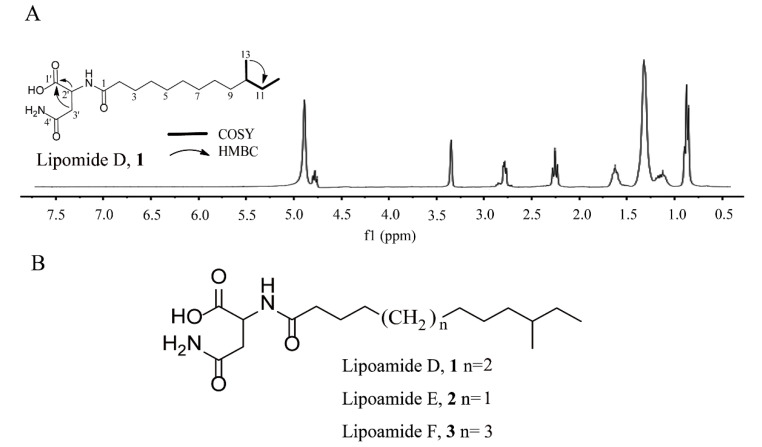
(**A**) ^1^H NMR analysis, key ^1^H–^1^H COSY, HMBC correlations, and (**B**) structure of lipomide D–F, **1**–**3**.

**Figure 2 molecules-26-01892-f002:**
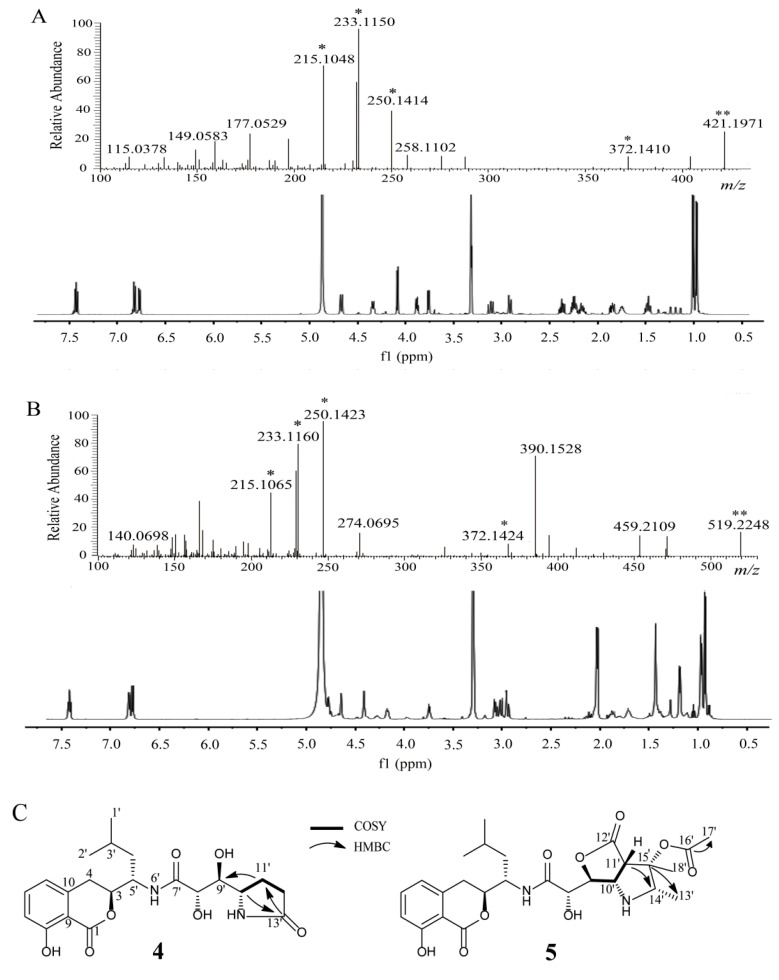
(**A**) HRMS/MS and ^1^H NMR analysis of amicoumacin D, **4** (**B**) HRMS/MS and (**B**)^1^H NMR analysis of amicoumacin E, **5**. (**C**) Key ^1^H–^1^H COSY, HMBC, relative stereochemistry of amicoumacin D, **4**, E, **5**. * Characteristic ion peak. ** Precursor ion peak.

**Figure 3 molecules-26-01892-f003:**
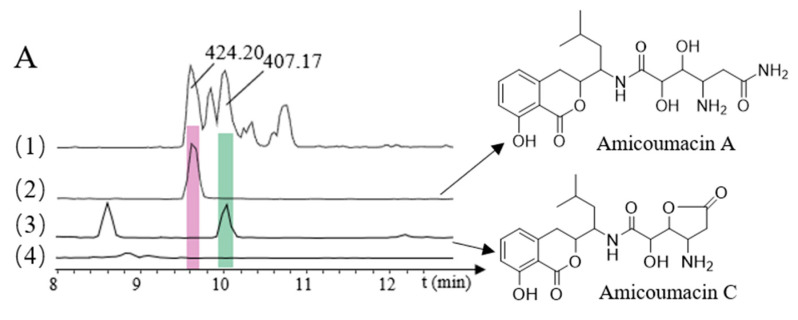
HPLC-MS analyses of heterologously produced amicoumacins. (**A**): (1) *B. subtilis* fmb60, (All MS); (2) Amicoumacin A (*m/z* 424.20) (3) Amicoumacin C (*m/z* 407.17); (4) *E. coli* GB05-MtaA, blank; (**B**): (1) *B. subtilis* fmb60, (All MS); (2) Bacilosarcin A (*m/z* 492.20); (3) *N*-butanonyl-amicoumacin C (*m/z* 477.20); (4) *E. coli* GB05-MtaA, blank.

**Figure 4 molecules-26-01892-f004:**
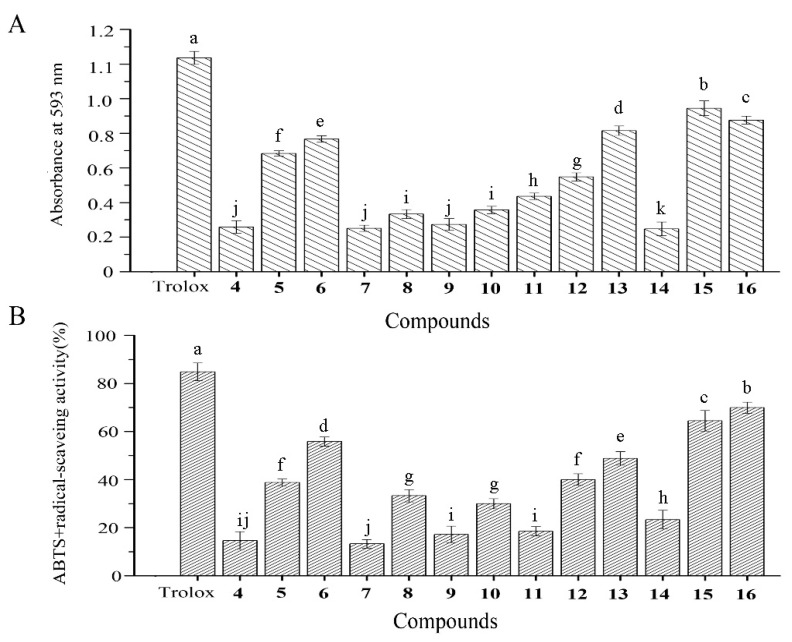
(**A**) FRAP activity and (**B**) ABTS^+^ radical-scavenging activity of the isolated compounds. Amicoumacin D, **4**, E, **5**, N-butanonyl-amicoumacin C, **6**, amicoumacin A–C, **7**–**9**, bacilosarcin A–C, **10**–**12**, *N*-acetylamicoumacin B–C, **13**–**14**, AI-77-H, **15**, lipoamicoumacins B, **16**. Results labeled with different lowercase letters (a–k) within each column indicate significant differences (*p* < 0.05).

**Table 1 molecules-26-01892-t001:** ^1^H NMR (*J* in Hz), ^13^C NMR spectral data of lipomide D–F, **1**–**3**, in methanol-*d*_4_.

Position	Lipomide D, 1		Position	Lipomide E, 2		Position	Lipomide F, 3	
	δ_H_ ppm (*J*, Hz)	δ_C_ ppm		δ_H_ ppm (*J*, Hz)	δ_C_ ppm		δ_H_ ppm (*J*, Hz)	δ_C_ ppm
Fatty acid								
1		176.2 C	1		176.1 C	1		176.2 C
2	2.23, t(2H,7.80)	37.0 CH2	2	2.23, m(2H)	37.0 CH_2_	2	2.23, m(2H)	37.0 CH_2_
3	1.61, t(2H,5.40)	26.9 CH2	3	1.61, m(2H)	26.9 CH_2_	3	1.61, m(2H)	26.9 CH_2_
4–7	1.31, m(10H)	28.3–31.2 CH2	4–8	1.31, m(8H)	28.3–31.2 CH_2_	4–9	1.32, m(12H)	28.3–31.2 CH_2_
8	1.15, m(1H)	30.7 CH2	9	1.15, m(1H)	30.7 CH_2_	10	1.15, m(1H)	30.7 CH_2_
	1.31, m(2H)			1.31, m(2H)			1.31, m(2H)	
9	1.31, m(1H)	35.8 CH	10	1.31, m(1H)	35.8 CH	11	1.31, m(1H)	35.8 CH
10	1.09, m(1H)	37.9 CH2	11	1.09, m(1H)	37.9 CH_2_	12	1.09, m(1H)	37.9 CH_2_
	1.31, m(1H)			1.31, m(1H)			1.31, m(1H)	
11	0.85, m(3H)	11.8 CH3	12	0.85, m(3H)	11.8 CH_3_	13	0.87, m(3H)	11.8 CH_3_
12	0.85, m(3H)	19.7 CH3	13	0.85, m(3H)	19.7 CH_3_	14	0.87, m(3H)	19.7 CH_3_
Asparagine								
1′		175.1 C	1′		175.2 C	1′		175.4 C
2′	4.70, t(1H,6.60)	50.5 CH	2′	4.70, t(1H,6.60)	50.5 CH	2′	4.70, m(1H)	50.5 CH
3′	2.76, m(1H)	37.9 CH2	3′	2.76, m(1H)	37.8 CH_2_	3′	2.82, m(1H)	37.4 CH_2_
	2.72, m(1H)			2.72, m(1H)			2.76, m(1H)	
4′		174.1 C	4′		174.2 C	4′		174.4 C

**Table 2 molecules-26-01892-t002:** ^1^H NMR (*J* in Hz), ^13^C NMR spectral data of amicoumacin D, **4**, and E, **5**, in methanol-*d_4_*.

Position	Amicoumacin D, 4			Position	Amicoumacin E, 5		
	δ_H_ ppm (*J*, Hz)	δ_C_ ppm	HMBC		δ_H_ ppm (*J*, Hz)	δ_C_ ppm	HMBC
1		171.1 C		1		171.3 C	
3	4.66, dt(1H,12.5, 3.0)	82.8 CH	4, 4′	3	4.80, m(1H)	82.1 CH	4
4	3.09,m(1H)	30.9 CH_2_	5	4	3.04, m(1H)	31.0 CH_2_	5
	2.90, dd(1H,16.5, 3.0)				2.95, dd(1H, 13.5, 2.5)		
5	6.77, d(1H, 7.0)	119.5 CH	4, 7	5	6.80, d(1H, 6.5)	119.8 CH	4, 7
6	7.46, t(1H,8.0)	137.5 CH		6	7.45, t(1H, 6.5)	137.5 CH	5
7	6.82, d(1H, 8.5)	116.7 CH	5	7	6.83, d(1H, 7.0)	116.7 CH	5
8		163.2 C	6, 7	8		163.3 C	6, 7
9		109.4 C	4, 5, 7	9		109.7 C	4, 5, 7
10		141.6 C	3, 4, 5, 6	10		141.7 C	4, 5, 6
1′	0.94, d(3H, 6.5)	22.0 CH_3_	2′, 3′, 4′	1′	0.93, d(3H, 5.5)	21.8 CH_3_	2′, 3′, 4′
2′	0.98, d(3H, 6.5)	23.7 CH_3_	1′, 3′, 4′	2′	0.98, d(3H, 6.0)	23.8 CH_3_	1′, 3′, 4′
3′	1.72, m(1H)	25.8 CH	1′, 2′, 4′	3′	1.73, m(1H)	25.7 CH	1′, 2′, 4′, 5′
4′	1.81,m(1H)	40.9 CH_2_	1′, 2′, 3′, 5′, 3	4′	1.83, m(1H)	40.1 CH_2_	1′, 2′, 3′, 5′
	1.47, m(1H)				1.46, m(1H)		
5′	4.33, dt(1H,10.5, 3.5)	50.4 CH	3′, 4	5′	4.19, m(1H)	51.6 CH	3′, 4
7′		175.2 C	5′, 8′, 9′	7′		172.7 C	5′, 8′, 9′
8′	4.09, d(1H, 6.5)	74.2 CH	9′, 10′	8′	4.43, m(1H)	74.4 CH	10′
9′	3.76, dd(1H, 4.5, 1.5)	75.2 CH	8′, 11′	9′	4.66, m(1H)	87.7 CH	8′, 10′
10′	3.88, m(1H)	57.8 CH	8′, 9′, 11′, 12′	10′	4.80, m(1H)	59.7 CH	14′
11′	2.22, m(1H)	22.7 CH_2_	9′, 10′, 12′	11′	3.09, d(1H, 6.5)	51.9 CH	10′, 14′
	2.16, m(1H)			12′		177.4 C	9′, 10′, 11′
12′	2.36, m(1H)	31.1 CH_2_	11′	13′	1.19, d(3H, 5.5)	17.1 CH_3_	14′
	2.23, m(1H)			14′	3.76, dd(1H,5.5, 11.0)	64.6 CH	11′, 13′, 18′
13′		181.5 C	10′, 11′, 12′	15′		78.1 C	11′, 13′, 14′, 18′
				16′		172.1 C	17′
				17′	2.05, s(3H)	22.3 CH_3_	16′
				18′	1.44, s(3H)	30.2 CH_3_	11′, 14′

**Table 3 molecules-26-01892-t003:** MIC values of lipomide D–F, **1**–**3**, amicoumacin A, **7**, and bacilosarcin B, **11**, toward selected microorganisms (µg/mL).

Microorganism	Erythromycin Gluceptate	Lipomide D	Lipomide E	Lipomide F	Amicoumacin A	Bacilosarcin B
*S. aureus* ATCC 25923	0.10	12.5	12.5	12.5	1.56	1.56
*M. luteus* CMCC 28001	0.39	25	25	25	1.56	1.56
*B. pumilus* CMCC 63202	1.56	12.5	12.5	12.5	3.13	1.56
*B. cereus* ATCC 14579	3.12	6.25	6.25	6.25	1.56	1.56
*L. monocytogenes* CICC 21662	0.39	12.5	12.5	12.5	1.56	3.13
*S. aureus* MRSA	1.56	12.5	12.5	12.5	6.25	3.13
*P. fluorescens* ATCC 49642	0.78	>100	>100	>100	6.25	6.25
*E. coli* ATCC 25922	50	>100	>100	>100	100	100

ATCC: American Type Culture Collection, CMCC: China Center of Medicine Culture Collection, CICC: China Center of Industrial Culture Collection. Multidrug-resistant *S. aureus* (MRSA) was isolated from Jiangsu Province Hospital.

## Data Availability

The authors declare that all data generated or analyzed during this study are included in this published article.

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
