# Peer review of "Genome Mining, Heterologous Expression, Antibacterial and Antioxidant Activities of Lipoamides and Amicoumacins from Compost-Associated Bacillus subtilis fmb60"

_molecules, 2021, doi:10.3390/molecules26071892_

Round 1

Reviewer 1 Report

The manuscript is well written and deals with current issues such as the search for new compounds with antibiotic activity, which provide a solution to the problems of resistance to traditional antibiotics. Also the use of genome minning is a current technique to discover new metabolites in new strains.
The study isolates and identifies new metabolites, belonging to the lipoamides and aminocoumacin families. In figures, the chemical shifts placed above of the RMN spectrum, as well as the integrals of the signals below, could be eliminated so that the figures would be clearer, as they are also repeated with these data in the supplementary material.
The new compounds that have been assigned have only relative stereochemistry, so it would be good to indicate this in the drawings of the structures in which they are already presented with defined chiral centers.
I miss a scheme where all the isolated compounds are presented, both the new ones and those already described previously in the literature for the strain.
The gene clusters were cloned in E. coli but fewer quantities and fewer compounds from the same family were obtained, which may be explained a little more in the work. On the other hand, good reasons are given for the biosynthesis pathway of the main B. subtilis metabolites thanks to cloning experiments.

And the work is completed with the antioxidant and antibacterial tests of the isolated compounds with moderately good results. 

Check the bibliography in the text, since 26 and 4 appear in two moments as superscripts. And in the supplementary material, in the part of the NMR spectra, it would not be bad to see the structures of the compounds above in some of the corners of the spectra.

Author Response

We are very grateful to the Editor and reviewers in recognising our work. We acknowledge all the comments and suggestions made by the Editor and reviewers that are very valuable in improving the quality of our manuscript. We have made all the required corrections properly as suggested by the Editor and reviewers. The corrections and amendments have been indicated in a different colour in the revised manuscript for easy follow-up.

Reviewer 2 Report

  1. Information about the whole metabolites profile of B.subtilis (Bs) should be added in the introduction (briefly). The Antimicrobial and antioxidants activity of the metabolites should be clarified in Intro.
  2. I was extremely uncomfortable working with combined results and discussion. It is quite difficult to split collected data and discussions other papers. I recommend dividing the Result and Discussion section into section results and section discussion.
  3. Table 3 (section 2.6) and Fig 4 (section 2.7). There is no information about statistical differences!
  4. The authors need to clarify number of replicates for each experiment in section 3.11. For example, if the analysis of metabolites has been done in triplicate it means that SE etc. should be presented in all tables and figures.
  5. I recommend adding in the title information that the antimicrobial activity of the Bs compounds has been studied.

Author Response

(The authors gave the same response as above.)

Round 2

Reviewer 2 Report

All comments have been addressed.